# Characterizing Ethereum Upgradable Smart Contracts and Their Security Implications

## Abstract

Upgradeable smart contracts (USCs) have been widely adopted to enable modifying deployed smart contracts. While USCs bring great flexibility to developers, improper usage might introduce new security issues, potentially allowing attackers to hijack USCs and their users. In this paper, we conduct a large-scale measurement study to characterize USCs and their security implications in the wild. We summarize five commonly used USC patterns and develop a tool, USCDetector, to identify USCs without needing source code. Particularly, USCDetector collects various information such as bytecode and transaction information to construct upgrade chains for USCs and disclose potentially vulnerable ones. We evaluate USCDetector using verified smart contracts (i.e., with source code) as ground truth and show that USCDetector can achieve an overall 96.26% accuracy. We then use USCDetector to conduct a large-scale study on Ethereum, covering a total of 60,251,064 smart contracts. USCDetecor constructs 10,218 upgrade chains and discloses multiple real-world USCs with potential security issues.

## 1  Introduction

Smart contracts are critical building blocks for decentralized applications (DApps) such as decentralized finance (DeFi) [68] and NFT [31]. As of Sep 2023, more than 61 million smart contracts have been deployed on Ethereum [50], the largest blockchain supporting smart contracts. To enforce transparency and trust decentralization, Ethereum and many other contract-supporting blockchains adopt the immutable smart contract design. That is, a smart contract, once deployed, cannot be changed or upgraded by any centralized entities. However, immutability conflicts with various legitimate causes to upgrade a deployed smart contract, such as introducing new functional features or patching security vulnerabilities, leading to inconvenience in practice. Thus, since 2016, various design patterns for upgradable smart contracts (USCs) have been introduced on Ethereum [5, 9, 12, 13], and widely adopted by DApps [62, 81]. Also, many third-party libraries, such as OpenZeppelin [7], have been developed to ease and accelerate USC development and deployment.

Despite all these efforts, developing USCs is still challenging and requires developers to be trained with security awareness [1]. Otherwise, security vulnerabilities might exist in USCs, allowing attackers to hijack USCs and further affect their users. Unfortunately, such vulnerabilities are not rare among USCs. For example, a widely adopted OpenZeppelin USC template is vulnerable to permanent state impairment that can potentially cause huge financial loss (e.g., over

$50m) [3]. Attackers have even successfully destroyed a USC and obtained all its ETH [10]. With more contracts integrating upgrade features, it becomes more likely that attackers target these upgradeable contracts in the future [11].

In this paper, we conduct a large-scale measurement study to characterize USCs and their security implications in the wild. We first introduce five commonly used USC patterns and their implementations. Specifically, our works cover the straightforward method (e.g., to deploy a new contract and migrate states), the Ethereum built-in method (i.e., Metamorphic contract), and three methods that decouple a contract into two sub-contracts (e.g., one immutable contract and one contract that can be modified). In addition, we present a series of security risks that could potentially cause serious consequences. Some vulnerabilities might enable off-path attackers to completely destroy target USCs (e.g., deny their service) and even hijack existing contracts. Other issues might put smart contract users into dangerous situations, such as losing assets or trading deprecated tokens. To the best of our knowledge, we are the first to systematically investigate several USC security issues on a large scale.

We develop a tool, USCDetector, to identify USCs and their security issues. Unlike previous work relying on source code analysis [53] and can only detect limited USC types (e.g., proxy-based [53]), USCDetector collects various information such as bytecode and transaction information, which are available for all contracts, to detect five types of USCs, construct their upgrade chains, and disclose potentially vulnerable ones. Thus, USCDetector can cover both unverified (i.e., without source code) and verified smart contracts. We evaluate USCDetector using a subset of verified smart contracts, and show that it can achieve an overall 96.26% accuracy.

We adopt USCDetector on Ethereum, covering a total of 60,251,064 smart contracts. USCDetecor constructs 10,218 upgrade chains with 91,956 USCs identified. Our results show some interesting observations: while the proxy-based approach is the most popular one, many developers attempt to mix different approaches to implementing USCs, which can offer more flexibility and facilitate batch processing. Moreover, USCDetector successfully discloses multiple real-world USCs with potential security issues. For example, we have identified 27 USCs lacking restrictive checks on the upgradable functions, potentially enabling anyone to hijack them. Additionally, we have discovered 117 vulnerable contracts that may completely disable USCs (e.g., become unusable forever). We have also identified tokens of many deprecated contracts are still listed in various decentralized exchanges, affecting many unaware users.

## 2 Background

### 2.1 Ethereum Blockchain

Blockchain is a public database that records transactions across many nodes in a decentralized network. Ethereum is a blockchain embedded with a single canonical computer, referred to as the Ethereum Virtual Machine (EVM) [40], maintaining states that everyone on the Ethereum network agrees on. There are two types of accounts on Ethereum: (1) externally owned account (EOA) controlled by anyone with private keys, and (2) contract account controlled by code (i.e., smart contract) [39]. Only an EOA can initiate a transaction, which can be used to transfer ETH or change EVM states. Typically, a transaction contains various information, including a unique identifier for the transaction (*transaction hash*), the sender address (*from*), and the receiver address (*to*). The *input* is the data sent along with the transaction. Transactions are verified and added to the blockchain based on cryptographic mechanisms, and typically cannot be changed without altering all subsequent blocks.

### 2.2 Ethereum Smart Contracts

Ethereum supports the execution of smart contracts, which essentially are computer programs that can automatically execute on EVM. Every smart contract contains a collection of code (i.e., functions) and data (i.e., states) that resides at a specific address. A smart contract can be written by *Solidity* [74] and then compiled to bytecode, which is the final code deployed on Ethereum and executed by EVM. EVM bytecode includes approximately 70 different opcodes for computations and communication with the underlying blockchain. Each opcode has two representations, a hex value and a more readable mnemonic. Specifically, the source code of smart contracts may not always be available, but the bytecode is publicly accessible on the blockchain [83].

Smart contracts can be created or executed with transactions. Once a contract is deployed, data can be stored in storage or memory for the function's execution. The functions in the deployed contract can get/set data in response to incoming transactions. Typically, *internal* functions can only be accessed within the contract or its derived contracts, while *external* functions can be called from other contracts/users [36]. To interact with an external function, the function needs to be identified by the first four bytes of the data sent with a transaction. These first four bytes are called *function selectors*, which are calculated by the function name and type of parameters through Keccak hash (SHA3).

Smart contracts serve as the foundation of decentralized applications (DApps) [41] such as Decentralized Exchanges (DEXs) [38]. DEXs are open marketplaces for exchanging tokens, which can represent different assets in Ethereum (e.g., lottery tickets, a fiat currency) [35]. Typically a smart contract that implements the ERC-20 standard (a standard for fungible tokens) [46] is referred to as a token contract.

### 2.3 Smart Contracts Immutability

Smart contracts are immutable once they are deployed on Ethereum. One reason is that Ethereum does not provide a built-in way to modify deployed smart contracts until the Constantinople hardfork [37]. Also, transactions for interacting smart contracts are packed into blocks to form the blockchain, which is difficult to modify. While the immutability of smart contracts achieves better transparency and security in general, there are certain problematic cases. First, it is difficult to fix vulnerable contracts, whose vulnerabilities may be either caused by their own logic problems or introduced by the programming languages (e.g., *Solidity* [73]). Second, it is difficult to add new features if the current smart contract can no longer meet the needs of users. As a result, several smart contract upgrading methods have been utilized to meet the increasing need for upgrading smart contracts.

## 3 Upgrading Patterns and Implementations

Smart contract upgrading is to modify the code executed in an address while preserving the contract's states. Understanding the implementation details of upgradeable smart contracts (USCs) is necessary to investigate their potential security threats. This section demystifies five commonly used upgrading patterns and their implementations.

### 3.1 Contract Migration

The idea of contract migration is to deploy a new smart contract with modified code (i.e., new version), which has an empty state, then migrate all states (e.g., data) from the old (i.e., old version) to the new one. Meanwhile, since the new version has a new address, other smart contracts interacting with the old version must also update to the new address, as presented in Figure 1a. From smart contract users' perspectives, after the migration, the new version contains all users' states (e.g., balances and addresses). Thus, users also need to switch to using the new address. For example, if the upgraded contract is a token contract, the old version should be discarded on the exchanges (e.g., Uniswap [81]), and the new version needs to be listed.

Typically, the deployer needs to make an official announcement claiming that the old version has been migrated to the new one. On Etherscan (an Ethereum blockchain explorer) [45], only once the deployer provides such information, Etherscan will label the old contract as "Old Contract". Since the new version and the old version are completely independent contracts, it is difficult to detect *contract migration* contracts. Bandara et al. [18] conducted a study on this pattern, by searching multiple keywords such as "token", "smart contract", "migration" to locate web pages related to blockchain and DApp migration. Then they manually analyzed selected web pages to obtain such migration pattern.

### 3.2 Data Separation

*Data Separation* is to split a contract into two sub-contracts, with one contract including logic (e.g., code) that can be

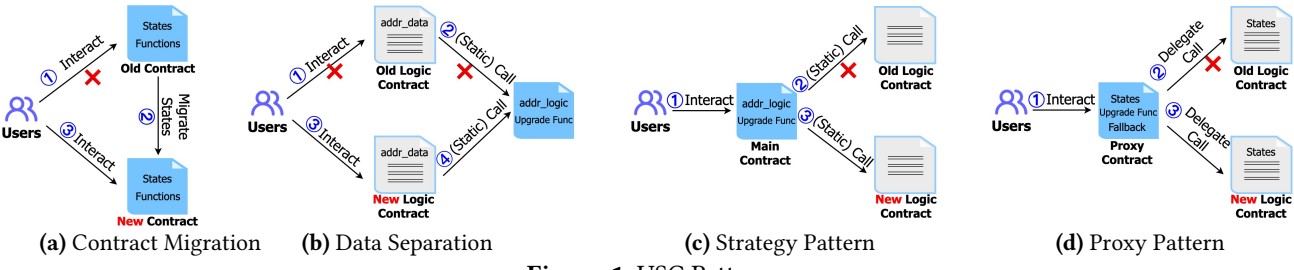

(a) Contract Migration    (b) Data Separation    (c) Strategy Pattern    (d) Proxy Pattern

**Figure 1.** USC Patterns.

modified later and the other contract preserves states. Users interact with the logic contract, which includes the address of the data contract and uses opcodes *CALL* or *STATICCALL* to interact with it for requiring data. The difference is that *STATICCALL* does not allow the called function to change state on EVM. Meanwhile, the data contract needs to configure the logic contract's address in its own state. Each time a smart contract wants to modify states, the data contract checks whether the caller contract matches its preserved logic address to protect states from being tampered with.

To upgrade, developers simply need to deploy a new logic contract, and then update the address in the data contract's storage. As all states are stored in the data contract, there is no need to do extra operations (e.g., state migration). Since users interact with the logic contract, after the upgrade, users also need to switch to the new logic contract (Figure 1b).

### 3.3 Strategy Pattern

*Strategy Pattern* also divides the original smart contract into separate ones: main contract and satellite contracts (i.e., logic contracts). The main contract includes both the core business logic and states, as well as the address of the logic contract. Similar to data separation, the main contract also interacts with the logic contract to execute certain functions using opcodes *CALL* or *STATICCALL* [42].

To upgrade, developers can deploy a new logic contract, and then update the new logic contract's address in the main contract. Since the main contract preserves all states, there is no need to migrate states during the upgrade process (Figure 1c). Compared with data separation, the advantage is that users and other smart contracts remain interacting with the main contract and do not need to switch addresses.

### 3.4 Proxy Pattern

Similar to data separation, *proxy pattern* keeps business logic and data in separate contracts (Figure 1d). However, in the proxy pattern, users interact with the storage contract (i.e., *proxy*), which preserves states including the address of the logic contract. The proxy contract delegates function calls to the logic contract using the opcode *DELEGATECALL*, which allows the proxy to call the logic contract, while the actual code execution happens in the context of the proxy. This means the proxy reads and writes to its own storage. It executes logic (e.g., functions) stored at the logic contract similar to calling internal functions [42].

In Solidity, a fallback function is executed if the called function does not match any functions in the proxy. The proxy can rewrite a custom fallback function that uses *DELEGATECALL* to delegate all unsupported function calls to the logic contract. While the proxy is immutable, the address of the logic contract can be replaced with the address of a new logic contract (e.g., upgrading [42]). Since the proxy reads and writes to its storage using the logic stored in the logic contract, the function for updating the logic contract's address can be placed either in the proxy or logic contract. In particular, if the function for updating the logic contract's address is in the logic contract, this pattern is called *universal upgradeable proxy standard* (UUPS) [13, 66].

To perform an upgrade, developers can deploy a new logic contract with modified code, and then update the new logic contract's address in the proxy contract. Since the proxy contract preserves all states, there is no need to migrate states during the upgrade process.

**Third-party Templates.** There are many third-party templates for implementing proxy-based USCs. For instance, OpenZeppelin [7] is a popular open-source framework providing upgradeable contract templates for developers, such as *UUPSUpgeadeable* [64] and *OwnableUpgradeable* [63].

### 3.5 Mix Pattern

Developers can mix the features of both strategy pattern and proxy pattern. It can directly call certain functions from the logic contract (via *CALL* or *STATICCALL*), and also delegate calls to the logic contracts (via *DELEGATECALL*).

### 3.6 Metamorphic Contract

On 2/28/2019, Ethereum performed Constantinople hardfork [37] starting from block 7,280,000, and introduced a new opcode *CREATE2* [21], which can be used for upgrading. It enables developers to deploy different codes to the same address. Developers need to utilize the *SELFDESTRUCT* opcode to wipe out the code and states of that address, then use *CREATE2* to redeploy code. In this way, the bytecode can be changed at the same address (e.g., upgrading). This contract is referred to as a Metamorphic contract [66]. However, this pattern has a drawback that it cannot preserve states after upgrading, as it must destroy the old contract first.

## 4 Potential Security Issues

This section presents several potential security issues in existing USCs, which might be exploited by independent

attackers (e.g., not the developers of target USCs) to hijack USCs. Some might become serious bugs affecting USC users.

### 4.1 Missing Restrictive Checks

USCs must implement restrictive checks on the upgrade functions ensuring that only contract admin can upgrade contracts. Otherwise, anyone can initiate contract upgrading and change the existing logic address to an arbitrary address. Attackers can even hijack the contract by changing the logic contract to one controlled by them.

For Metamorphic-based USCs, the first step is to call a function to destroy the Metamorphic contract by using the opcode *SELFDESTRUCT*. Similarly, if this function has no restrictive checks, anyone can potentially destroy this contract. Furthermore, once the old contract is destroyed, attackers can hijack the contract by redeploying a new bytecode (e.g., following the Metamorphic-based approach).

### 4.2 Insufficient Restrictive Implementation

For a proxy-based USC, while users should interact with the proxy, malicious users can send transactions to the logic contract directly. In general, this does not pose a threat, since the state of the logic contract does not affect the proxy. However, it will become a serious issue, if a malicious user becomes the owner of the logic contract (by initializing it), and then destroys it by calling a function containing *SELFDE-STRUCT*. The proxy will delegate all calls to a self-destroyed logic contract, causing a denial of service (DoS). Particularly, this issue happens if the deployer has only initialized the proxy, but ignored to initialize the logic contract. The proxy contract utilizes *DELEGATECALL* to call the initialize function of the logic contract, which runs in the context of the proxy contract, thus leaving the logic contract uninitialized.

There are two cases to destroy the logic contract after being the owner of the logic contract. For case I, the logic contract includes a function containing *SELFDESTRUCT*. A malicious user can simply call this function to destroy the logic contract. For case II, the logic contract has a function containing *DELEGATECALL*. Then it can delegate a call to a predefined function containing *SELEDESTRUCT*, which can be called to destroy the logic contract. Particularly, case II mainly exists in UUPS and can cause devastating consequences. For example, the UUPS template provided by OpenZeppelin [7] includes a function upgradeToAndCall(), which introduces such a problem [3]). For UUPS-based USCs, if the logic contract is destroyed, the USC becomes unavailable forever, as the upgrade function lies in the logic contract.

### 4.3 Missing Checks on Logic Address

The upgrade function needs to conduct necessary logic checks on the contract address of the target logic contract. Otherwise, an upgrade may replace the existing logic contract with an arbitrary address, potentially causing irreversible consequences. For example, if an upgrade sets the new logic address to an EOA address while the upgrade function is in the logic contract (e.g., UUPS), it can make the *USC*

completely unusable. We enumerate several issues that are likely to occur with logic addresses as follows.

- **External owned address.** The new address is an EOA. When a proxy delegates a call to an EOA, no function will be executed since EOA has no state or code. Thus, this call will always return success, essentially causing a DoS.
- **Empty contract.** The new address is a contract but without any states or functions. Similarly, it causes DoS issues.
- **Zero address.** The new address is zero (DoS threat).
- **Same address.** The same upgrade transaction was submitted multiple times, resulting in duplicate upgrades. Such upgrades might cause a waste of gas.
- **Non-upgradeable logic address in UUPS.** The new address is zero, or an EOA, or does not contain a function setter. It will make USCs unavailable forever.

### 4.4 Contract Version Issues

For *contract migration* and *data separation*, the old (logic) contract is replaced by a new contract, which interacts with users. Thus, after migration, the old contract should be deprecated. For example, Etherscan can publish an announcement indicating that the old version is no longer in use.

**Old contract still in use after migration.** This problem is that, if the old version is not self-destructed, users may not know the upgrade and still interact with the old contract, which might cause severe security issues. For example, if the old contract has security vulnerabilities, users' assets in this contract might be in danger.

**Token contracts on DEXs.** If the newly deployed contract is a token-based contract, it is important to collaborate with DEXs [38] to ensure that the new contract will be listed and the previous one will be discarded. Otherwise, DEXs might still list the old tokens, and future exchange users will swap (e.g., trade) with the old tokens.

## 5 Methodology

We design an analysis tool, USCDetector, to characterize USCs and investigate their potential vulnerabilities in the existing Ethereum blockchain. Unlike previous works [53] that only detect proxy-based USCs based on source code analysis, USCDetector identifies multiple types of USCs based on bytecodes, which are always publicly accessible, and thus enable us to characterize both verified and unverified USCs in the wild. The high-level idea is that developers attempt to use common keywords (e.g., upgrade, update) in their upgrading functions [53], which are available in bytecodes. With the help of other collectable information (e.g., transactions), we can identify USCs with high accuracy.

The overall workflow is illustrated in Figure 2. USCDetector first disassembles smart contract bytecodes into opcodes and extracts function selectors. Then it filters a list of USC candidates based on a set of pre-defined common upgrade function keywords. Additionally, USCDetector collects various supplementary information such as logic contract addresses and transaction information. Finally, based on all

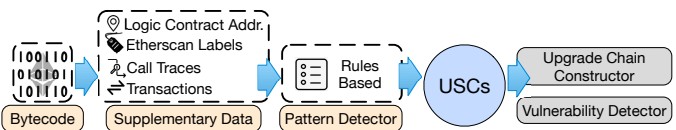

**Figure 2.** Overview of USCDetector.

collected information, a rule-based pattern detector identifies USCs and their upgrading patterns.

### 5.1 Bytecode Collector

The bytecode collector first utilizes an Ethereum RPC (Remote Procedure Call) service [23] to collect smart contract bytecode. Then we disassemble bytecodes into opcodes using an npm package *truffle-code-utils* [80], and further utilize *abi-decode-functions* [16] to extract functions selectors. Particularly, a smart contract's bytecode first compares the function selector in the transaction's input to all function selectors in the smart contract, and then jumps to the matching function for execution. *abi-decode-functions* extracts function selectors based on such patterns (i.e., match and jump) using pre-defined templates. However, we find that it only covers a subset of templates so some function selectors might be missed. We thus extend it to include more common patterns.

The above method can extract most function selectors from the immutable contract of USCs (e.g., proxy contract in the proxy pattern, main contract in the strategy pattern, and data contract in the data separation). However, it cannot extract function selectors of the logic contract in the *strategy pattern*. Particularly, the main contract can use the *PUSH4* opcode to push a 4-byte function selector onto the stack, and then use the opcode *CALL* to call that function whose logic is implemented in the logic contract. In other words, the function selector included in the main contract actually represents a function in the logic contract. We also extend the *abi-decode-functions* tool to handle such cases.

Finally, the bytecode collector also extracts various opcodes (e.g., *SELFDESTRUCT*) and other function related information (e.g., fallback function) for pattern detection (Section 5.4). Table 4 in Appendix A lists the detailed description.

### 5.2 Common Upgrade Function Keywords.

Our idea is to construct and apply a set of keywords that are commonly used in upgrade functions to filter function selectors. Obviously, the quality of such a dataset is critical for USC detection accuracy. We first analyze the Ethereum mainnet dataset [53], which contains 2,295 unique (3,822 in total) proxy-based USCs. From a total of 9,842 upgrade functions, we extract 111 unique upgrade functions, and find most of them can be divided into groups containing five keywords: *set* (570), *upgrade* (8,548), *update* (229), *change* (88), and *replace* (22). We then query these keywords on the *Ethereum Signature Database* [6], which contains 4-byte signatures of functions in EVM. The Ethereum signature database returns all functions including selected keywords and their corresponding 4-byte signatures. However, functions containing keywords cannot ensure they are upgrade functions.

For example, `setUserData(address,uint256,uint256)` is clearly not used for upgrading. Thus, we only keep the functions containing meaningful and related words, such as *contract*, *implementation*, and *logic*, etc. We also include some function names that we manually collect online, such as *enableModule* from *Gnosis Safe Contracts* [49].

### 5.3 Supplementary Data Collector.

We further collect various information in addition to bytecode to assist USC detection.

**Logic Contract Collector.** In proxy-based USCs, particularly UUPS, the upgrade function exists in the logic contract, instead of the proxy contract. We then utilize an RPC (*evm-proxy-detection*) to collect the logic contract's address of smart contracts that contain *DELEGATECALL* opcode (e.g., potential proxy-based USCs). Such addresses are further fed to the *Bytecode Collector* to process their bytecode.

**Call Trace Collector.** Metamorphic-based USCs need to destroy the old contract (i.e., using the *SELFDESTRUCT* opcode) and then redeploy new bytecode using the *CREATE2* opcode to that address. Thus, to identify metamorphic-based USCs, we need to collect the call traces of the transaction that creates a new contract, and then detect if the contract is indeed created by *CREATE2* in that call traces. Particularly, we first request Etherscan API [44] to obtain the creators and transaction hashes of smart contracts that contain *SELFDESTRUCT*. Then we use transaction hashes to request Openchain API [61] to obtain call traces. Finally, we parse the opcodes and input them into the detector.

**Etherscan Crawler.** We also crawl various information from Etherscan websites for all contracts that are labeled as "Old Contract". As mentioned in Section 3.1, the "Old Contract" is labeled by Etherscan when the deployer provides the addresses of both the old and new contracts and a link to an official announcement regarding the contract migration [22]. The crawler then collects all related information.

**Transaction Analyzer.** We further collect all transactions whose *input* contains upgrade functions (which are potential USCs' upgrade transactions). Specifically, we collect them by querying Bigquery [28] that uses *ethereum-etl* [19] to extract data from the Ethereum blockchain every day. Then we use *ethereum-input-data-decoder* [43] to decode the transactions' *input* and extract function selectors and arguments.

### 5.4 Rule-based Pattern Detector

Based on the characteristics of different USC patterns, we develop a rule-based pattern detector to identify them. The detailed rules and notations are listed in Table 5 in Appendix A. Specifically, the *proxy pattern* must have (1) both *DELEGATECALL* and the *fallback* function exist in the proxy contract; and (2) an upgrade function in either the proxy or logic contract (i.e., UUPS). We first use three criteria for detecting *strategy pattern* and *data separation*: (1) the existence of upgrade functions; (2) *CALL* or *STATICCALL* used; and (3) particular external functions. We further distinguish

**Table 1.** USCDetector Accuracy on Randomly Sampled Data.

| Patterns | TP | FP | Accuracy |
|---|---|---|---|
| Proxy Pattern | 244 | 6 | 97.6% |
| Data Separation | 43 | 2 | 95.55% |
| Strategy Pattern | 70 | 2 | 97.22% |
| Strategy or Data Combined | 421 | 19 | 95.68% |
| Mix Pattern | 50 | 3 | 94.3% |
| Metamorphic Contract | 7 | 0 | 100% |
| Total | 722 | 28 | 96.26% |

these two patterns using transaction information: strategy-based USCs call logic contracts, while data separation USCs are called by logic contracts. For *Metamorphic* contracts, we check whether *SELFDESTRUCT* exists and *CREATE2* is in the call trace. If contracts meet the rules for both proxy-based and strategy-based, we mark them as mix patterns. Finally, it is difficult to identify the *contract migration* pattern at the bytecode level. Therefore, we select all contracts labeled with the "Old Contract" label on Etherscan and remove contracts that are identified as other patterns (e.g., data separation).

### 5.5 Upgrade Chain Constructor

Finally, we construct contract upgrade chains for USCs that have already performed upgrades. The upgrade chain for *contract migration* is straightforward: we simply concatenate contracts based on Etherscan labels. For other patterns, to perform an upgrade, an EOA must initiate a transaction to the contract that needs an upgrade. We thus rely on collected transaction information to build the chain. For example, upgrading metamorphic contracts is to redeploy new bytecode on the same address. We then chain detected metamorphic contracts with the same address. With the upgrade chain, we further check multiple security issues (details in Section 7).

### 5.6 USCDetector Evaluation

The Smart Contract Sanctuary project [65] is a project including verified Ethereum smart contracts on Etherscan. As of March 22, 2023, this dataset contains 320,080 verified smart contracts, with source code available. We input this list into USCDetector and have identified 8,653 USCs, with 2,517 proxy-based USCs, 7 Metamorphic contracts, and 568 mix pattern USCs. In addition, there are 5,746 USCs using *data separation* or *strategy pattern*. Among them, we further utilize transaction information to separate them. Since not all USCs have performed upgrades (i.e., have upgrade transactions), we detect 468 *data separation* and 988 *strategy pattern* USCs.

To evaluate the accuracy, we randomly select and manually verify 750 smart contracts including all patterns based on their source code and decompiled code. The result of random sampling is listed in Table 1. Overall, we are able to achieve 96.26% accuracy, with only a few false positives. Particularly, it is worth noting that our list of upgrade functions is not obtained from this dataset, indicating that our methodology has the potential to accurately detect unknown USCs without their source code.

## 6 Measurement in the Wild

We utilize USCDetector to detect smart contracts collected from Bigquery [50], which has exported 60,251,064 smart contracts (date: 6/5/2023) from the Ethereum blockchain. We first group smart contracts based on their bytecode, so that smart contracts in each group have identical bytecode. We take the smart contract with the earliest creation time from each group as the representative for further analysis.

In the total of 964,585 groups, we have identified 27,420 groups, with 1,936,657 individual USCs (column "Raw" in Table 2). We find that proxy-based pattern dominates existing upgradable methods, with 1,866,904 USCs from 4,964 groups. However, most of them are dominated by one group, namely *OwnableDelegateProxy* [8], which includes 1,546,462 USCs. This group was created by the smart contract *WyvernProxyRegistry* [14], which is maintained by OpenSea [62], a popular NFT market. *WyvernProxyRegistry* creates a proxy contract for each seller on OpenSea (i.e., the seller owns the contract) for executing sellers' actions. Obviously, the main purpose is not for upgrading; we thus exclude such contracts in our following analysis (as column "Number" in Table 2).
**Basic Characterization.** Table 2 shows the detailed breakdown of each pattern. We also present the aggregated ETH and transaction volumes. The most popular is the proxy pattern, with 43,650 contracts containing more than 736k ETH and 51M transactions. One possible reason is the wide adoption of third-party templates (e.g., Openzeppelin [7]), which provide open source contract libraries for developing smart contracts. Figure 3 presents the number of different patterns of USCs over time. The blue line indicates that data separation and strategy pattern were the main upgrade methods until the proxy method (green) was introduced. The Metamorphic contract comes after February 2019, and is not popular due to its drawback mentioned in Section 3.
**Upgrade Chains.** In total, we have constructed 4,692 upgrade chains for proxy-based USCs, 4,337 for strategy-based or data separation patterns, 201 for mix pattern, 110 for metamorphic-based USCs, and 878 for contract migration. Figure 4 shows the CDF of upgrade chains. Most of them have conducted less than 20 upgrades, while the longest chain contains 92 upgrades. Also, 89.1% of the contract migration approach have only 1 upgrade, which is reasonable as contract migration essentially is to deploy a new contract.

We also find that many upgrades are conducted by different owners (i.e., owner change): there are 185 proxy-based, 97 strategy/data separation, and 52 mix-pattern USCs.
**Mix Pattern Demystified.** We detect 23,725 (Raw) mix-based USCs, which combine features of both strategy and proxy patterns. We find that there are different ways to implement the mix pattern. One popular way (20,591) utilizes an upgradeable intermediate contract. The main (proxy) contract can *CALL* the intermediate contract to get the return address for the logic contract. Then, the main contract can

**Table 2.** USC Breakdown.

| Patterns | Raw | Number | ETH | Trans. |
|---|---|---|---|---|
| Proxy Pattern | 1,866,904 | 43,650 | 736K | 51M |
| Data Separation | 1,024 | 1,024 | 131 | 1M |
| Strategy Pattern | 2,444 | 2,444 | 280K | 7.7M |
| Strategy or Data Combined | 44,017 | 42,808 | 287K | 20M |
| Mix Pattern | 23,725 | 1,420 | 2.8K | 5.7M |
| Metamorphic Contract | 3,097 | 3,097 | 68.5 | 3.3M |
| Contract Migration | 984 | 984 | 410 | 16M |

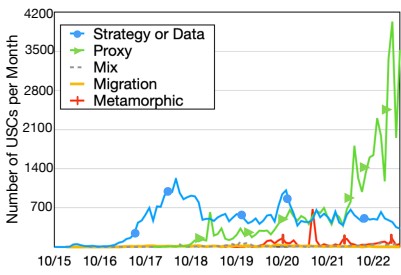

**Figure 3.** Patterns Over Time.

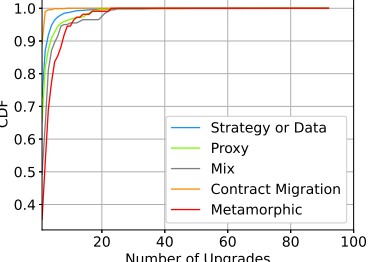

**Figure 4.** Upgrade Chain CDF.

delegate calls to the logic contract using the returned address. In this way, the main contract does not preserve the actual address of the logic contract. When performing an upgrade, developers simply modify the logic contract's address in the intermediate contract. It does not require any operations on the main contract for upgrading. Thus, it enables uniform upgrades across multiple USCs that share a logic contract.

Another popular approach (704) is similar: the main contract preservers the logic contract's address, and also contains the upgrade function. When upgrade, the main contract first *CALL* the logic contract to see if a new logic address is returned. If yes, the main contract updates the new logic address, and then delegate the call to the new one.

**Hierarchy Upgrade.** Interestingly, we find some developers utilize strategy-based USCs to further upgrade multiple proxy-based USCs. For example, a proxy-based USC already contains an upgrade function (in the proxy contract) that can upgrade its logic contracts. Then, the developers utilize a strategy-based main contract to directly *CALL* the upgrade function in the proxy contract to upgrade its logic contract. The advantage is that, developers can utilize one main contract to manage multiple proxy contracts. USCDetector finds 1,349 such strategy-based USCs, and 2,628 proxy-based USCs were upgraded in this way.

## 7 Security Issue Characterizations

This section presents the security issues related to upgrades. We first introduce the methods for identifying them. We also manually verify them based on decompiled code.

### 7.1 Missing Restrictive Checks

If a USC misses restrictive checks, an unauthorized user might be able to upgrade this contract. Based on our collected upgrade chains, we extract USCs that were upgraded by multiple different owners, and further keep USCs if there is an owner who only upgrades the contract once. We then manually check their upgrade functions' decompiled code to confirm the missing of restrictive checks.

For metamorphic-based USCs, we check the upgrade chains of metamorphic contracts and locate functions containing *SELFDESTRUCT* from decompiled code. Then we check if these functions have or miss restrictive checks.

**Results.** We find that the issue exists in mix pattern (2 USCs), Strategy Pattern (2 USCs), and Metamorphic Contract (23 USCs). List 1 in Appendix A presents a real-world

example derived from DApp LANDProxy [32]. Additionally, we find that many transactions attempt but fail to upgrade USCs: these upgrading transactions are not initialized by the owner(s) performing upgrades. Although these transactions fail due to restrictive checks, they indicate that attackers potentially have started to hijack vulnerable USCs.

### 7.2 Insufficient Restrictive Implementation.

To identify potentially vulnerable logic contracts in proxy-based USCs, we first extract logic contracts that contain *SELFDESTRUCT* for case I, and UUPS-based logic contracts for case II. Then, we query the states of these logic contracts on Oko [33], which is an Ethereum explorer listing states of all smart contracts. We mark USCs that do not have any state as potentially vulnerable, as it indicates that these logic contracts have not performed any initialization.

Finally, we manually verify whether there is an initialization function that can declare contract ownership in a logic contract from their decompiled code. Specifically, for case I, if there is such a function in a logic contract, this logic contract is vulnerable as it also includes *SELFDESTRUCT*. For case II, we consider a logic contract is vulnerable if it contains the function `upgradeToAndCall` and can be called directly (OpenZeppelin has disabled this function to be called through active proxy after UUPS template version 4.3.2 [4]).

**Results.** For case I, we find 1 vulnerable UUPS-based USC and 51 vulnerable normal proxy-based USCs (i.e., the upgrade function lies in the proxy). For case II, we detect 66 vulnerable USCs. In total, these vulnerable contracts own $6,350.09 assets (e.g., ETH and tokens), and 12 USCs still have recent transactions on 9/2023. Note that vulnerabilities in UUPS can completely disable USC (i.e., its proxy contract becomes unusable forever). Listings 2 and 3 in Appendix A present vulnerable UUPS-based examples of both cases.

### 7.3 Missing Checks on Logic Address

We utilize the addressing information from upgrade chains to detect logic address issues (e.g., same address, empty contract, EOA, and zero address). Particularly, for empty contracts and EOA, we can not automatically distinguish them. Thus, we manually check the lists with a history of upgrades to non-contracts to classify them.

For detecting non-upgradeable logic addresses in UUPS, we collect all logic addresses from our identified UUPS list, and input them to the *Pattern Detector* again without enabling

**Table 3.** Detected Logic Issues.

| Patterns | Same Addr. | Zero Addr. | EOA | Empty Contract | Non-Upgrade Addr. in UUPS |
|---|---|---|---|---|---|
| Proxy Pattern | 160 | 9 | 55 | 3 | 215 |
| Data Separation | 16 | 1 | 3 | 0 | 0 |
| Strategy Pattern | 87 | 111 | 20 | 0 | 0 |
| Strategy or Data Combined | 122 | 128 | 51 | 0 | 0 |
| Mix Pattern | 22 | 2 | 2 | 0 | 0 |
| Total | 304 | 139 | 108 | 3 | 215 |

rule detection. We then filter logic addresses that contain no upgrade function and conduct manual verification.

**Results.** We find that these issues are prevalent across all patterns except Metamorphic contracts. Table 3 presents the detailed number. The most common problem is successive upgrades to the same address, with a total number of 304. These consecutive transactions are often separated by only a few seconds, suggesting that it might be the same upgrade but committed multiple times, causing a waste of gas. Also, zero addresses, EOA, and empty contracts are quite common. Particularly, as mentioned in Section 4, setting the logic address to a non-contract address does not affect other functions in the strategy pattern and data separation. Instead, in the proxy and mix patterns, this can cause denial-of-service threats. Figure 5 in Appendix A presents a real-world example that the logic address is an EOA.

We further explore whether these USCs have corrected their logical addresses later. We find that 35 proxies' logic contracts are still EOA accounts. Even worse, the logic addresses of 15 proxy contracts are actually set to phish accounts (flagged as "Phish" by Etherscan) and 12 of them still have transactions. Additionally, one proxy has encountered a malicious transaction from a phishing account, so the creator has set the logic address to zero. One proxy's logic address points to a contract that contains no functions or states.

### 7.4 Contract Version Issues

**Old contract still in use after migration.** From our constructed upgrade chains of *migration-based* USCs, we extract their upgrading reasons from Etherscan announcements. As some contracts might use both old and new versions, we only target contracts that explicitly mention that the old version is no longer in use. Then, we utilize Etherscan to observe if there are still transactions after the new contracts have been created. For *data separation*, we check the usage of old logic contracts after a new logic contract is deployed.

For *contract migration*, we find that 16 old contracts explicitly state that they are no longer in use. However, 10 of them are still interacted with users after publishing their migration announcements, generating a total of 908 transactions. For *data separation*, we detect 21 USCs that have performed upgrades, but users are still interacting with their old logic contracts, generating a total of 253 transactions. These results demonstrate that *contract migration* and *data separation* are not trivial: users might continue interacting with the old contracts after a successful upgrade.

**Token contracts on DEXs.** Token lists [2] is a community-led new standard for creating ERC20 token lists, containing lists from many DEXs like Uniswap and CoinMarketCap [29]. From the token list, we pick lists of tokens that have been updated recently, and utilize upgrade chains to check whether old contracts are still listed in these token lists.

Among 878 constructed upgrade chains (*contract migration*), 11 have both old and new token contracts listed in at least one DEX; 45 have their old and new token contracts listed in different DEXs; 50 have their old token contracts listed in one of the DEXs, but without listing their new token contracts, indicating that old token contracts have not been replaced yet. Particularly, one old token contract has 21,275 holders, much more than its new token contract (only 3,794 holders). The results indicate that many old tokens are still listed on DEXs, and thus users may swap these old tokens.

## 8 Related Works

**Smart Contract Upgrades.** USCs have attracted many research efforts [17, 58, 70] on understanding their characteristics. For example, *Rodler et al.* designed EVMPatch [70] to rewrite the bytecode of exploited smart contracts and deploy them as upgradeable proxy contracts. Our work focuses on characterizing existing mainstream smart contract upgrading patterns and their security implications.

One closely related work is Proxy Hunting [53], which focuses on security issues of proxy-based USCs from contracts' source code. Instead, our work covers more USCs patterns (e.g., data separation and strategy pattern) on unverified smart contracts (i.e., without source code). We also investigate several security issues that have not been studied by previous literature. Finally, there are two works on detecting metamorphic contracts using opcode *CREATE2* [15, 48]. Our work further utilizes transaction call traces to check whether *CREATE2* is used to create smart contracts.

**Smart Contract Security Analysis.** Extensive research efforts have been conducted on analyzing various security issues (e.g., reentrancy vulnerabilities) of smart contracts [20, 30, 60, 67, 69, 72, 85] at both the source code level [24, 34, 47, 51, 52, 54, 56, 59, 75–77, 79, 82, 84] and bytecode level [25–27, 55, 57, 70, 71, 78, 86]. For example, *Wang et al.* developed NPChecker [82] to detect nondeterministic vulnerabilities in smart contracts. DefectChecker [25] is introduced to detect smart contracts' defects from bytecode. Different from previous efforts, our work focuses on uncovering existing USCs that have security issues at the bytecode level.

## 9 Conclusion

This paper presents a large-scale measurement study on USCs and their potential security issues. We have developed USCDetector to identify five types of USC without needing source code, with an accuracy of 96.2%. We have used USCDetector to analyze 60,251,064 smart contracts, and discovered multiple real-world USCs that have potential security issues.

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

**Table 4.** Opcodes and Function Information Extracted From Bytecode.

| Information | Abbreviation | Descriptions |
|---|---|---|
| CALL | CALL | Call a method in another contract. |
| DELEGATECALL | DCALL | Call a method in another contract using the storage of current contract. |
| STATICCALL | SCALL | Call a method in another contract without state changes. |
| SELFDESTRUCT | SDES | Destroy the contract. |
| Fallback | FBK | Fallback function will be called when a non-existent function is called on the current contract. |
| Functions | {FUNC} | A set of functions belong to current contract. |
| Other Functions | {OFUNC} | A set of functions that current contract calls another contract. |

**Table 5.** The Rules Specified for Each Pattern.

| Pattern | Rules |
|---|---|
| Proxy Pattern | $UpgradeFunction \in (\{FUNC\}_{proxy} \vee \{FUNC\}_{logic}) \wedge DCALL \wedge FBK$ |
| Strategy Pattern | $UpgradeFunction \in \{FUNC\}_{main} \wedge (CALL \vee SCALL) \wedge \{OFUNC\}_{main}$ |
| Data Separation | $UpgradeFunction \in \{FUNC\}_{data} \vee Rule_{strategy}$ |
| Mix Pattern | $Rule_{proxy} \wedge Rule_{strategy}$ |
| Metamorphic Contract | $SDES \wedge CREATE2 \in CallTrace_{tx}$ |
| Contract Migration | $Addr_{old} \wedge Announcement \wedge Addr_{new}$ |

# A   Appendix
## I. USCDetector Details
### Notations.

| | |
|---|---|
| $\{FUNC\}_{proxy}$ | : a set of functions belong to proxy. |
| $\{OFUNC\}_{main}$ | : a set of functions that the main contract calls the logic contract. |
| $CallTrace_{tx}$ | : the call traces of the transaction. |
| $Rule_{strategy}$ | : the rule for detecting strategy pattern. |
| $Addr_{old}$ | : the address of the old contract. |

### Rules.

**Proxy Pattern Rule** requires *1) *DELEGATECALL*, (2)*fall-back* function, and (3) the upgrade function must exist either in the proxy or logic contract.

**Strategy Pattern Rule** requires three elements to exist in the main contract at the same time. These elements are the upgrade function, *CALL* or *STATICCALL* (or both of them), and the functions that the main calls the logic contract.

**Data Separation Rule** requires the upgrade function to exist in the data contract. If the data contract calls functions of other contracts, it then has the same features as the strategy pattern.

**Mix Pattern Rule** requires that both the proxy's rule and strategy's rule must be satisfied.

**Metamorphic Contract Rule** requires *SELFDESTRUCT* to exist in the contract, while the call trace of the transaction that creates the contract must contain *CREATE2*.

**Contract Migration Rule** requires the address of the old contract, and announcement, and the address of the new contract must exist in the records we obtain from Etherscan at the same time.

## II. Additional Listings of Vulnerable Examples

List 1 presents a simplified version of a Mix Pattern contract that has no restrictive check on contract admin, derived from a real-world DApp LANDProxy [32]. There is no admin check on the *upgrade* function, and thus anyone can overwrite the logic address (Line 3).

```
1  contract Proxy {
2      function upgrade(IApplication newContract,
       bytes data) public {
3          currentContract = newContract;
4          newContract.initialize(data);
5      }
6      function () payable public {
7          require(currentContract != 0);
8          delegatedFwd(currentContract, msg.data);
9      }
10     ...
11 }
```

**Listing 1.** A simplified version of LANDProxy

Listing 2 presents a case I example. It lists the decompiled code of a simplified version of a (unverified) vulnerable UUPS-based logic contract. There is a function `initialize()` (Line 8) that can be called directly the only check in this function is about the state stored in *stor0*, which denotes if this function has been called (Line 14 - Line 20). Since this contract has no state, its ownership can be obtained by calling the initialize function. Attackers can further call the `destruct()` function to destroy this contract, disabling its proxy contract.

```
1  def storage:
2    owner is address at storage 151
3      ...
4  def destruct(address to):
5    if owner != caller:
6        revert 'not the owner'
7    selfdestruct(to)
8  def initialize():
9    owner = caller
10   ...
```

**Listing 2.** Decompiled code of a Logic Contract that Contains SELFDESTRUCT

Listing 3 presents a case II example: a simplified version of a UUPS-based logic contract. It does not contain *SELF-DESTRUCT* opcode, but can still be destroyed through the function `upgradeToAndCall`. The upgrade mechanism is implemented correctly for the proxy: only the owner can perform upgrades (Line 10). However, the `upgradeToAndCall` function can be directly called by the logic contract owner. An attacker can call the function `initialize()` (Line 3) to take ownership of the logic contract, and further destroy it by calling the function `upgradeToAndCall()` (Line 10).

Particularly, the `_upgradeToAndCallSecure()` function includes a rollback test to validate that the new logic address also has an upgrade function (Line 29). However, this test can be bypassed by performing twice upgrades: first by resetting the *rollbackTesting* value (Line 28) and then by calling `_upgradeToAndCallSecure()` function to a function containing *SELFDESTRUCT* in the upgrading logic contract (Line 26). This would destroy the logic contract and cause the proxy's *DELEGATECALL* to point to a self-destructed logic contract.

```
1  contract Token is ..., UUPSUpgradeable{
2      ...
3      function initialize(string memory _name,
       string memory _symbol) initializer public {
4          ...
5      }
6      ...
7  }
8  abstract contract UUPSUpgradeable is
       ERC1967UpgradeUpgradeable {
9      ...
10     function upgradeToAndCall(address
       newImplementation, bytes memory data) external
       payable virtual {
11         _authorizeUpgrade(newImplementation);
12         _upgradeToAndCallSecure(newImplementation,
       data, true);
13     }
14     ...
15 }
16 abstract contract ERC1967UpgradeUpgradeable {
17     ...
18     function _upgradeToAndCallSecure(
19         address newImplementation,
20         bytes memory data,
21         bool forceCall
22     ) internal {
23         ...
24         _setImplementation(newImplementation);
25         if (data.length > 0 || forceCall) {
26             _functionDelegateCall(
       newImplementation, data);
27         }
28         BooleanSlot storage rollbackTesting =
       getBooleanSlot(_ROLLBACK_SLOT);
29         if (!rollbackTesting.value) {
30             ...
31             _upgradeTo(newImplementation);
32         }
33     }
34     ...
35 }
36 ...
```

**Listing 3.** Decompiled code of a Logic Contract that Function Setter Can Be Called Directly

## III. Additional Figures of Vulnerable Examples

Figure 5 presents a real-world example of proxy-based USC. Figure 5a shows a transaction from a proxy's admin to *initialize* the proxy contract. As the function *initialize()* is supposed to exist in the logic contract, the proxy contract simply delegates the call to the logic contract (Figure 5b). However, the logic address is actually an EOA. Thus, the *initialize()* was not executed and eventually has no state change.

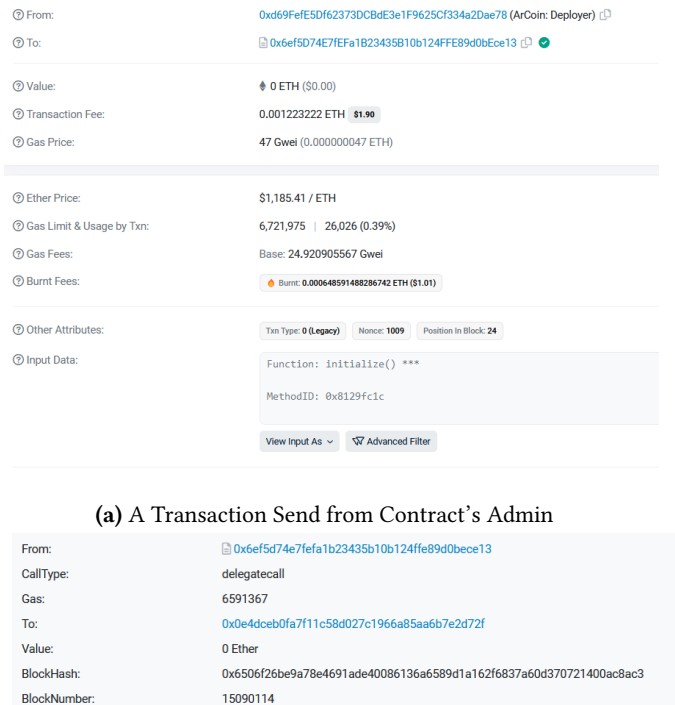

**(a)** A Transaction Send from Contract's Admin

**(b)** The Proxy Contract Delegate Call to an EOA

**Figure 5.** USC Patterns.

