# OpenReview forum: "Characterizing Ethereum Upgradable Smart Contracts and Their Security Implications"
_ACM.org/TheWebConf/2024/Conference — TheWebConf24_

### Official Review · Reviewer_pYP6 · 2023-10-25

**Novelty:** 4
**Technical Quality:** 4

**Review:**

## Pros

+ A large-scale measurement study
+ Some findings related to security issues

## Cons

- No comparison with state-of-the-art work, namely Proxy Hunting, in terms of results
- The significance of contracts with security issues is unclear
- It is unclear whether potential security issues are being reported to developers
- No evaluation of false negatives


## Details

This is a paper that targets an important problem of smart contract security, particularly a measurement study of upgradable contracts in the real world.  I think the paper's results are generally interesting and should be made available to the community.  At the same time, my main concerns are described below:

(1) The paper does not conduct an evaluation-wise comparison with prior work, namely Proxy Hunting (USENIX'23).  While I understand that the difference is closed-source vs. open-source and proxy-based vs. other types, it would be interesting to see how many contract issues that your approach can discover but Proxy Hunting cannot.  I checked the Proxy Hunting paper and their source code is available.  Therefore, I think a comparison is totally possible.  And I would encourage the authors to perform such a comparison.

(2) While the paper does discover some security issues of smart contracts, the significance of these security issues is unclear.  First of all, it is unclear how many ethers/transactions are being affected.  Second, it is unclear whether those contracts are popular.  I think more statistics would be helpful to understand the impacts of the discoveries. Also, it is unclear how many of those contracts are open-source and how many are closed-source.  And it is also helpful to know how many of them are proxy-based, i.e., discoverable by Proxy Hunting.

(3) Following up on (2), I think you should report your findings to the developers of the contracts and try to obtain their feedback.  It would be nice to obtain acknowledgement from the developers.  Also, how many of these security issues are fixable?  If none of them are fixable, would it matter whether we discover them or not?

(4) I do not see any evaluation of false negatives.  It is unclear to me how many upgradable contracts are missed in the results. I would suggest that the authors curated a dataset (or maybe use the results from Proxy Hunting) to estimate the False Negatives of the proposed approach.

**Questions:**

1) What are your FNs?

2) How does your approach compare with Proxy Hunting in terms of results?

3) How significant are your results?

**Ethics Review Description:**

It is unclear whether the discovered security issues have been reported to the developers.

**Ethics Review Flag:**

Yes

**Reviewer Confidence:**

3: The reviewer is confident but not certain that the evaluation is correct

**Scope:**

3: The work is somewhat relevant to the Web and to the track, and is of narrow interest to a sub-community

---

### Official Review · Reviewer_XM5d · 2023-11-21

**Novelty:** 5
**Technical Quality:** 4

**Review:**

Rebuttal response
----------------------------
After reading your rebuttal, some of my worries have been assuaged. While I still have some reservations, I have (slightly) increased my grade on technical quality - based on the assumption you will follow through on the promises made in the rebuttal to provide more details.

Summary
----------------------------
The paper investigates upgradable smart contracts. To this end, an analysis tool is designed and applied. It finds thousands of upgradable smart contracts, identifying vulnerabilities in multiples



Main contributions
----------------------------
- method for determining upgrade chains
- method for determining whether a given smart contract is upgradeable
- application of these methods to over 60M contracts
- analysis of results


Strong points
----------------------------
- Relevant
- interesting


Weak points
----------------------------
- Weak targeting
  E.g., sec 3, 4: these sections lack specifics (e.g., code listings, code
  examples, discussions of features offered by the EVM to facilitate proxying
  and to facilitate caller verification, etc.)
  Similarly: sec 2.3 and the lede of section 3 aren't really necessary for
  readers already somewhat (remotely) familiar with smart contracts.
- Somewhat ad hoc underpinning of methodology
  Approach is based on keywords. The way these keywords are chosen is somewhat
  haphazard. Keywords are derived from a small set of preselected smart
  contracts, which gives rise to false positives, which are then eliminated by
  further keywords whose origin is not detailed.
- Insufficient details
  see comments for athors
  Short synopsis: The description in the paper is far from sufficient to
  replicate the implementation's functionality. It focuses too much on
  ideas, not enough on specifics.
  Case in point: it is insufficiently clear how to detect an upgradeable
  smart contract, let alone how to characterise one as being of a
  specific type.
  (One can infer methods for doing so, the methods the paper used are
  not clear.)


Key review questions
----------------------------
- Is the concept scientifically publishable?
	Yes.

- Are the ethical aspects sufficiently considered? (esp. informed consent)
	Not clear if there was any attempt at responsible disclosure for the
	found vulnerable smart contracts.
	(I am aware that it is frequently hard to impossible to know who
	authored a smart contract; nevertheless, *some* effort is required.)

- Is the current form publishable?
  That is: writing + execution crosses threshold of non-rejectability
  (does writing and/or execution not undermine the scientific contributions)
  Also not overstating to such an extent that it is verging on lying
	No.
	The ideas definitely merit publication; the writeup is too unclear about
	details of methodology (e.g., "various information" is collected). This
	needs to be addressed before the paper can be accepted.
	Note that the lack of checking for false negatives is not per se
	required, but would be such a trivial addition (in sec 5.6, using
	the Etherscan data set) that it is recommended.

- Is it "good enough" for the current conference?
	It fits well enough -- the CfP did mention smart contracts.
	Note: I would feel this would be much more suited to a separate
	"blockchain & smart contracts" track than a web security track.



Overall evaluation
----------------------------
I like the premise of the paper. Unfortunately, the paper falls short of that premise, in my view. It lacks too many details on how to identify the various types of contracts, it does not show code examples at all, and it does not discuss in sufficient depth what facilities the EVM offers for determining caller identity and how delegating calls work.



Comments for authors
----------------------------
- pg 1, "we have identified 27 USCs lacking restrictive checks..."
  You should mention what steps you took for responsible disclosure.
- pg 2, sec 2.3, sec 3's lede
  I think the point about upgrading smart contracts was sufficiently clear
  following the intro. Also, "demystifies" is rather grandstanding.
- Sec 3 and 4 overall:
  These are rather introductory and light on details.
  While the concepts might become clear, a reader would not have learned
  anything that would help in detecting specific forms of pgradeable contracts.
- Sec 3, overall:
  This section needs some tightening up. You probably want to first clarify
  several features of the EVM (CALL/STATICCALL/DELEGATECALL) and be clear
  about where smart contract data resides. For describing each pattern,
  it would be good to more clearly state where the logic resides, where the
  data resides, which contract is called by users, and which contract is
  updateable.
  Comments on individual patterns below.
  o  Sec 3.1
     This section is light on necessary implementation details.
     You should at least explain how kill functions work, i.e., what facilities
     the EVM offers to kill a contract, delegate a call, and what is known about
     the caller. I'm not sure if an explanation about gas is warranted, but
     it's a bit weird to talk about Ethereum without mentioning it (and you
     do mention gas later, but don't explain it).
  o  Sec 3.2
     This description is hard to fully grok. The discussion of CALL/STATICCALL
     comes out of nowhere; highlighting the need for an earlier description of
     EVM functions.
     Secondly, it is not readily apparent what the benefit of this pattern is
     over contract migration - burden on users seems same, burden on developers
     seems increased. (Perhaps this is related to being hard to grok?)
     Furthermore, I would recommend adding a simple example (that is, source
     code).
  o  Sec 3.3 vs 3.4
     The difference between "strategy" and "proxy" is insufficiently
     clear -- also in Fig. 1. It seems to hinge on where the data resides and
     on CALL/STATICCALL vs. DELEGATECALL.
  o  Sec 3.5, mixing
     I can understand that this is possible, but it is not at all clear why
     on earth anyone would want to do that.
     Some words on that would be very welcome.
  o  Sec 3.6
     +  SELFDESTRUCT opcode not explained
     +  unclear if new bytecode can take less/more space than previous contract.
     +  example code would elucidate this pattern greatly
     +  "it cannot preserve states" -- only true for locally held states.
        (This goes again to the main question of "where is data kept", which
        is insufficiently clear throughout this section.)
     +  not shown in Fig. 1
- Sec 4.1
  It is not made clear how to implement restrictive checks. An example
  is needed -- e.g., a description of the facilities the EVM offers.
- Sec 4.1, "Furthermore...":
  Please elaborate.
- Sec 4.2, "There are two cases":
  This is poorly worded; please clarify.
  I think you mean that after becoming owner, the malicious user can
  call a function containing SELFDESTRUCT either directly
  (CALL/STATICCALL), or indirectly, by calling a function
  containing a SELFDESTRUCT. For an indirect call, it is important
  that the call is a DELEGATECALL, otherwise the wrong contract is
  obliterated.
- Sec 4.2, "UUPS"
  Please expand the acronym (it's been too long since its introduction).
- Sec 4.3
  List not needed. If you start pointing to the wrong thing, the right
  thing becomes unfindable.
  (Also: "gas" mentioned but never explained.)
- Sec 5, "unlike previous works [53]"
  Use singular, not plural
- Sec 5, "bytecodes" --> bytecode
  (Bytecode is uncountable)
- Sec 5, overall
  There is not quite enough detail here to give me confidence I'd be
  able to replicate the work accurately. There is enough of a
  description of the ideas so that I could do something similar, but the
  triggers for the various components of the USCDetector are insufficiently
  clear. Any other implementation based on this description would likely
  deviate significantly and lead to other results.
  That indicates a significant issue with the paper.
- Sec 5.1, "utilises" --> uses
  "utilise" can come across as an example of bad writing -- it does so for me.
- Sec 5.3:
  o  "we further collect various information"
     Please be more precise.
  o  "evm-proxy-detection"
     How does this Node.JS module not solve 95% of your contribution?
     More details are desperately needed.
  o  "We also crawl various information"
     Please be more precise
  o  "whose input contains upgrade functions"
     How do you determine the input contains upgrade functions?
- Sec 5.4
  This is the meat of the method. It shouldn't be relegated to an appendix,
  it should be presented -- and *defended* -- in the main paper.
- Table 1:
  consider advice from https://people.inf.ethz.ch/markusp/teaching/guides/guide-tables.pdf
  Main points: only 3 horizontal lines, no vertical lines.
- Sec 5.6
  With this dataset, it is very straightforward to estimate incidence of
  upgradeable smart contracts - and thereby estimate the false negative rate
  of your detector.
- Sec 6, "WyvernProxyRegistry"
  The way this is currently phrased calls into question the accuracy of
  Table 1.
  I think you're right to include these contracts in Table 1 and ignore
  them here, but not because "the main purpose is not for upgrading".
  That is not a quality measure you apply to *any* other contract, after
  all.
  So please motivate this better (and make clear why your false positive
  numbers aren't 1.5M out of 1.8M(!) ).
- Sec 6, "another popular approach (704)"
  You can't really call this popular if the first had a count of 20k+.
- Table 2, 3
  Again: 3 horizontal lines (above/below, and below headers), no
  vertical lines.
  Also: do not center numbers. Properly align them -- the 5 in 5.7M
  should be in the same horizontal place as the 1 in 1M and in 51M.
- Sec 8, smart contract security analysis:
  This is blatant padding of the reference list. Tighten up the writing
  in the earlier parts significantly to make space for an actual related
  work section.

**Questions:**

I don't really have a question - I believe I understand the paper well and that it is an interesting piece of research, but that it requires a significant rewriting effort to include sufficient details.

**Ethics Review Description:**

The paper found flaws in other people's code, but does not make any statement about responsible disclosure.

**Ethics Review Flag:**

Yes

**Reviewer Confidence:**

3: The reviewer is confident but not certain that the evaluation is correct

**Scope:**

3: The work is somewhat relevant to the Web and to the track, and is of narrow interest to a sub-community

---

### Official Review · Reviewer_ExvA · 2023-11-21

**Novelty:** 5
**Technical Quality:** 6

**Review:**

The authors characterize Ethereum upgradable smart contracts in five distinct categories and design USCDetector, a tool for identifying USCs without having access to their source code. By collecting various information, e.g., bytecode and transaction information, USCDetector is able to construct upgrade chains and achieves a high accuracy in identifying USCs. The authors leverage their tool to perform a large-scale study on Ethereum and uncover several implications due to insecure USCs and practices in the wild.

Although the paper falls outside my area of expertise, I liked reading it and seems to shed light on significant issues stemming from insecure practices in regards to deployed USCs. I would like to ask the authors whether they plan to open-source USCDetector's code so as to facilitate further research efforts in the area. I'm sure the rich metadata the tool collects, as well as the upgrade chains it constructs would be invaluable to other studies as well.

Regarding the related work on smart contract security, the authors mention that their work differs in that it focuses on uncovering security issues in existing USCs at the bytecode level. However, prior work [25,26,82] seems to also do the same (for different types of vulnerabilities). I think the paper should be better related to prior studies and explicitly highlight their key differences. In addition, the security evaluation relied on USCDetector's findings, but required significant manual effort; prior work [27,82] has demonstrated fully automated ways to identify vulnerabilities in smart contracts. Due to the potential severity of the demonstrated vulnerabilities/security issues, it would be really useful to automatically detect them. Is this something that could be done in the context of your work?

Finally, I would like to ask whether the authors considered any options in disclosing or remedying the identified issues. I assume that "regular" disclosure, as in other security related studies, might not be trivial due to Ethereum's anonymity. I think, however, that if a vulnerability stems from a given template, then that template's provider could be notified.

- Pros
    + Automatically characterized and detected deployed USCs according to their upgrade technique
    + Discovered security issues affecting deployed USCs

- Cons
    + No source code available
    + Difference from prior work should be more specific
    + Disclosure to affected parties (if possible)

**Questions:**

- Is USCDetector going to be publicly released?

- Could the security evaluation have been carried out automatically, e.g., by leveraging prior works' bytecode-level techniques?

- Were and can the discovered vulnerabilities/issues be disclosed to some affected parties?

**Reviewer Confidence:**

2: The reviewer is willing to defend the evaluation, but it is likely that the reviewer did not understand parts of the paper

**Scope:**

3: The work is somewhat relevant to the Web and to the track, and is of narrow interest to a sub-community

---

### Official Review · Reviewer_PkEb · 2023-11-22

**Novelty:** 5
**Technical Quality:** 6

**Review:**

The paper presents a security analysis of "Upgrade Smart Contracts", an Ethereum mechanism used to update smart contracts once deployed. From this analysis, the authors developed a SAST tool (called USCDetector) running on contracts' bytecode and present the scan results.

List of Pros:
- the paper is well written and clearly articulated
- the security analysis sheds light on new attack scenarios
- the tool works on information readily available (bytecode / transaction information), making it very usable

List of Cons:
- relevant bytecode is retrieved by looking for common keywords such as 'upgrade', meaning coverage won't be 100%. The approach to find true positives is however sound and quite comprehensive.
- unclear whether the contract owners were acknowledged of the issues

After rebuttal
- the authors answered all my questions and I am happy about the commitments they made to further enhance the manuscript.
- I would suggest acceptance for the paper

Minor:
- I noticed a typo ("SELEDESTRUCT" instead of "SELFDESTRUCT")

**Questions:**

- Did you warn contract owners (responsible disclosure), did they acknowledge the issue(s)?
- is there any improvement that could be made on the coverage side?

**Ethics Review Description:**

n.a.

**Reviewer Confidence:**

3: The reviewer is confident but not certain that the evaluation is correct

**Scope:**

4: The work is relevant to the Web and to the track, and is of broad interest to the community

---

### Official Review · Reviewer_sABi · 2023-11-24

**Novelty:** 5
**Technical Quality:** 4

**Review:**

# Paper Summary
This paper conducts a large-scale measurement study to characterize Ethereum upgradeable smart contracts (USCs) and their security implications. To this end, this work develops a tool named USCDetector to identify USCs, and manually analyzes the potential security issues. More specifically, USCDetector extracts various information such as the common upgrade function keywords in the bytecode of USCs, and then utilizes the designed rule-based pattern detector to identify five types of USCs. Based on the identified USCs and their upgrade chains, manual analysis finds four types of security issues, including missing restrictive checks, insufficient restrictive implementation, missing checks on logic addresses, and contract version issues.

# Strengths
+ a large-scale measurement study to characterize USCs and their security implications in the wild.
+ disclose multiple real-world USCs with potential security issues.
+ well-written presentation.

# Weaknesses
- lack of clear clarity of contributions
- limited depth in discussions and insights.
- the paper template is incorrect, but not fatal for the acceptance


# Detailed comments for authors:
This paper systematically investigates five types of Ethereum USCs and their four types of security implications on a large scale. The findings are important for understanding USCs' characteristics and potential security issues, and would be helpful for developers to improve the security of USCs. Overall, I appreciate the efforts of this work; however, the paper could be improved in the following aspects:

1. Adding clear clarity of the paper's contributions in Section I.

2. Adding more experiments to thoroughly evaluate USCDetector.

The main evaluation for USCDetector is about the accuracy in identifying USCs. The evaluation results in Table 1 show that the "accuracy" of USCDetector is 96.26%; however, the evaluation metric actually is "precision" (calculated by TP / (TP + FP)). The performance of USCDetector in terms of "recall" is also important.
Moreover, USCDetector utilizes two types of information for identification, including common upgrade function keywords from bytecodes and other collectable information such as transactions. It is important to evaluate the effectiveness of each of these types in the identification.

3. Adding more discussion and insight.

The paper provides detailed results regarding the identified USCs and their security issues. However, what insights can be derived from these results? For instance, which type of USC has higher security and would be suggested to be used? How to identify issues before an upgrade goes live?

Additionally, the paper can further discuss the limitations of this work and the potential threats to the conclusions.

**Questions:**

1. How many USC patterns are studied in this work? Section 3 mentions five commonly used patterns, but Sections 3.1 to 3.6 introduce a total of six patterns. Moreover, Table 5 also lists six patterns.

2. How is the number "1,936,657" calculated in the second paragraph of Section 6? The sum of the numbers in column "Raw" of Table 2 is 1,942,195. Additionally, how is the number "43,650" calculated in column "Number" of Table 2? By excluding 1,546,462 USCs that are not for upgrading, there remain 320,442 USCs (calculated as 1,866,904 - 1,546,462).

3. As stated in Section 6, there are a large number of contracts identified by USCDetector as USCs but actually not for upgrading. Does this observation imply a potential accuracy issue with USCDetector, which might contradict the conclusion drawn in Section 5.6?

**Reviewer Confidence:**

3: The reviewer is confident but not certain that the evaluation is correct

**Scope:**

4: The work is relevant to the Web and to the track, and is of broad interest to the community

---

### Decision · Program_Chairs · 2024-01-22

**Decision:**

Accept

**Comment:**

The reviewers appreciate the extensive measurement study reported in this paper and suggest several changes to further improve the study, that the authors promised to incorporate in their work during the discussion phase. Reviewers criticized the limited novelty with respect to prior work and some insufficiently explained methodological aspects. Some of these issues have been partially mitigated during the discussion phase.

 ---